# Global analysis of regulatory divergence in the evolution of mouse alternative polyadenylation

Mei-Sheng Xiao[1,†], Bin Zhang[1,2,†], Yi-Sheng Li[1], Qingsong Gao[1], Wei Sun[1,2] & Wei Chen[2,3,*] (ID)

## Abstract

Alternative polyadenylation (APA), which is regulated by both *cis*-elements and *trans*-factors, plays an important role in post-transcriptional regulation of eukaryotic gene expression. However, comparing to the extensively studied transcription and alternative splicing, the extent of APA divergence during evolution and the relative *cis*- and *trans*-contribution remain largely unexplored. To directly address these questions for the first time in mammals, by using deep sequencing-based methods, we measured APA divergence between C57BL/6J and SPRET/EiJ mouse strains as well as allele-specific APA pattern in their F1 hybrids. Among the 24,721 polyadenylation sites (pAs) from 7,271 genes expressing multiple pAs, we identified 3,747 pAs showing significant divergence between the two strains. After integrating the allele-specific data from F1 hybrids, we demonstrated that these events could be predominately attributed to *cis*-regulatory effects. Further systematic sequence analysis of the regions in proximity to *cis*-divergent pAs revealed that the local RNA secondary structure and a poly(U) tract in the upstream region could negatively modulate the pAs usage.

**Keywords** alternative polyadenylation; evolution; regulatory divergence
**Subject Categories** Chromatin, Epigenetics, Genomics & Functional Genomics; Genome-Scale & Integrative Biology; Transcription
**Mol Syst Biol. (2016) 12: 890**

## Introduction

The 3′ end cleavage and polyadenylation of nascent RNA is not only an essential step for the maturation of most mRNAs, but also serves as an important regulatory mechanism in controlling gene expression (Elkon *et al*, 2013; Tian & Manley, 2013). Recent genome-wide studies estimated that more than 70% of mammalian genes and about half of genes in flies, worms, and zebrafishes underwent alternative polyadenylation (APA), in which mRNA transcripts from the same gene locus could have multiple 3′ ends (Jan *et al*, 2011; Derti *et al*, 2012; Smibert *et al*, 2012; Ulitsky *et al*, 2012; Hoque *et al*,

2013). APA could change 3′ UTR, via usage of different polyadenylation sites (pAs) in the terminal exon, or in addition resulted in different protein isoforms with distinct properties, via the usage of pAs in the upstream exonic/intronic regions or the splicing of alterative last exon. The diverse 3′ UTRs generated by APA may contain different sets of *cis*-regulatory elements, thereby modulating the stability (Chen & Shyu, 1995; Barreau *et al*, 2005; Bartel, 2009; Jonas & Izaurralde, 2015), translation (de Moor *et al*, 2005; Lau *et al*, 2010), intracellular localization of mRNAs (Ephrussi *et al*, 1991; Bertrand *et al*, 1998; An *et al*, 2008; Niedner *et al*, 2014), or even the subcellular localization and function of the encoded proteins (Berkovits & Mayr, 2015). Often, APA shows tissue-specific pattern and is extensively regulated during development and upon stimuli. Whereas the global APA-mediated shortening of 3′ UTR was linked to cell proliferation, oncogenic transformation, pluripotency, lymphocyte activation, and neuronal stimulation, the 3′ UTR lengthening was observed during embryonic development and differentiation (Sandberg *et al*, 2008; Ji & Tian, 2009; Lau *et al*, 2010; Hoque *et al*, 2013; Miura *et al*, 2013; Gruber *et al*, 2014). Additionally, dysregulation of APA has also been observed in oncological, immunological, and neurological diseases (Gieselmann *et al*, 1989; Bennett *et al*, 2001; Masamha *et al*, 2014).

The pAs is defined by the interaction between multiple *cis*-elements in the nascent RNA transcripts and *trans*-acting RNA binding proteins. In mammals, the most well-known *cis*-element is the canonical AAUAAA hexamer and its close variants located 15 to 30 nucleotides upstream of the cleavage sites, which are defined as polyadenylation signal (PAS) and directly recognized by the cleavage and polyadenylation specificity factor (CPSF) subunits: CPSF30 and Wdr33 (Tian *et al*, 2005; Proudfoot, 2011; Chan *et al*, 2014; Schonemann *et al*, 2014). Other auxiliary motifs/elements include upstream UGUA motif bound by cleavage factor Im (CFIm) and downstream U-rich or GU-rich elements targeted by cleavage stimulation factor (CSF; Elkon *et al*, 2013). The outcome of APA is dependent on how efficiently each alternative PAS is recognized by these 3′ end processing machineries, which is under further regulation by additional *cis*- and *trans*-factors. It has been shown that several RNA binding proteins could enhance or repress the usage of distinct pAs through binding in their proximity (Brown & Gilmartin, 2003; Yao *et al*, 2012; Masamha *et al*, 2014; Gruber *et al*, 2016). However,

1   Laboratory for Functional Genomics and Systems Biology, Berlin Institute for Medical Systems Biology, Berlin, Germany
2   Department of Biology, Southern University of Science and Technology, Shenzhen, Guangdong, China
3   Medi-X Institute, SUSTech Academy for Advanced Interdisciplinary Studies, Southern University of Science and Technology, Shenzhen, Guangdong, China
    *Corresponding author. Tel: +86 755 88018449; E-mail: chenw@sustc.edu.cn
    †These authors contributed equally to this work

compared to alternative splicing, where numerous studies were devoted to the identification of the responsible *cis*-elements as well as *trans*-regulators, the study of APA regulation remains under-explored.

Changes of gene expression represent one major driving force underlying the evolution of phenotypic differences across different species (Hsieh *et al*, 2003; Rifkin *et al*, 2003; Nuzhdin *et al*, 2004; McManus *et al*, 2010). Given that the gene expression is a multi-step process, it is conceivable that evolutionary changes at each layer along this process could exert their distinct contribution. However likely due to the technical convenience, most of the genome-wide investigations in this field were focused on the study of transcriptional divergence. More recently, global aspect of alternative splicing, RNA decay as well as mRNA translation, and even protein stability started to be addressed. In comparison, the extent of APA divergence remains largely unexplored; particularly, how such divergence could arise from the change in *cis*-regulatory elements and/or *trans*-acting factors is totally unclear.

Here, to globally investigate the APA divergence in a mammalian system, we first comprehensively characterize the pAs expressed in the two mouse inbred strains (*Mus musculus* C57BL/6J and *Mus spretus* SPRET/EiJ) and then quantify their usage of the alternative pAs. Next, to address the relative contribution of *cis*- and *trans*-regulatory changes, we further analyzed the allele-specific APA pattern in their F1 hybrids. In F1 hybrids, the nascent RNA transcripts from both parental alleles are subject to the same *trans*-regulatory environments; thus, the observed differences in allele-specific pattern should only reflect the impact of *cis*-regulatory divergence. The contribution of *trans*-regulatory elements can then be inferred by comparing the allele-specific differences with the total APA differences between the parental strains. The two parental strains chosen in this study diverged ~1.5 million years ago, which results in ~35.4 million single nucleotide polymorphisms (SNPs) and ~4.5 million insertion and deletions (indels) between their genomes (Keane *et al*, 2011). Such a high sequence divergence allowed us to unambiguously determine the allelic origin for a large fraction of sequencing reads, thereby enabling accurate quantification of allelic APA for thousands of genes. In the fibroblasts derived from the two parental strains, we identified a total of 51,446 pAs, 75% of which were from 13,808 genes. Among the 24,721 pAs from 7,271 protein-coding genes expressing multiple pAs, 3,747 showed significant divergence between the two strains. By comparing them to allele-specific pattern in F1 hybrids, we could attribute such divergence predominately to *cis*-regulatory changes. Systematic sequence analysis of the regions in proximity to the *cis*-divergent pAs revealed that the local RNA secondary structure and a poly(U) tract in the upstream region could negatively affect the pAs usage.

# Results

### Construction of the pAs reference

To build a reference set of pAs clusters expressed in the fibroblasts derived from C57BL/6J and SPRET/EiJ mouse strains, we utilized the 3′ READS method, which has been developed to identify 3′ end of polyadenylated transcripts without use of oligo (dT) priming (Fig 1A; Materials and Methods). In total, we generated

two replicate 3′ READS datasets for each strain. On average, around 80% of the filtered reads could be uniquely mapped to the reference genome sequences, of which almost 40% harbored at least two non-genomic T at the 5′ end that were considered as pAs supporting (PASS) reads (Table EV1; Materials and Methods). Based on these PASS reads, a total of 38,218 and 38,836 pAs clusters were identified for C57BL/6J and SPRET/EiJ fibroblasts, respectively (Materials and Methods).

The nucleotide composition in the genomic region flanking the cleavage sites of identified pAs clusters resembled the pattern reported before (Hoque *et al*, 2013; Spies *et al*, 2013; Fig EV1A and B). Consistent with previous studies, more than 50% of the identified pAs clusters contained the canonical PAS motif AAUAAA within the region 100 nt upstream of the cleavage sites, whereas only < 10% were not associated with any known PAS variants (Fig 1B). In contrast, the frequency of these known PAS motifs was present only in 70 and 17% of the upstream region of pAs determined with the raw PASS reads (raw pAs) and reads without non-genomic T at 5′ end (pseudo pAs), respectively (Figs 1B and EV1C; Tian *et al*, 2005).

After filtering out pAs clusters that were identified in one strain, but could not be unambiguously mapped to the genome of the other strain, the remaining pAs from the two mouse strains were combined as a reference pAs set containing 51,446 pAs clusters (Fig EV1D–F; Materials and Methods). Based on Ensembl gene annotation, 75% (38,390) of these pAs clusters were assigned to 12,561 protein-coding genes and 1,247 non-coding gene, whereas the remaining 13,056 pAs were located in the inter-genic regions. Among the 38,390 genic pAs clusters, 13,406 were located within 50 nt away from the 3′ end of RNA transcripts annotated in Ensembl. Importantly, 80.6% of these pAs were almost identical (within a distance of 5 nt) to the annotated 3′ end, demonstrating the high quality of our data (Fig 1C). Among the 12,561 protein-coding genes, from which 36,191 pAs were derived, 66% had at least two pAs (Fig 1D). As shown in Fig 1E, we defined those overlapped with annotated gene end as Terminal pAs, other alternative pAs were classified into four types including Tandem 3′ UTR pAs, Alternative last exon pAs, Intronic pAs, and Internal exonic pAs (Fig 1E). In agreement with previous observation, the vast majority of these alternative pAs were located in the last exon resulted in tandem 3′ UTR (Fig 1F; Hoque *et al*, 2013).

### APA divergence between C57BL/6J and SPRET/EiJ

Although 3′ READS was highly effective in identifying pAs and could be used for quantification of the APA isoform expression, it was cumbersome and required a large amount of starting material. Therefore, in the following quantitative analysis on strain- or allele-specific pAs usage, we applied a simpler oligo-dT priming-based method (3′ mRNA-seq from Lexogen; Materials and Methods). Given the well-known mispriming issue, that is, oligo-dT could primer at internal A-rich sequences, to estimate the pAs usage, we only counted the 3′ mRNA-seq reads that were uniquely mapped and with their 5′ end within the reference pAs clusters identified by 3′ READS (Materials and Methods).

To explore the APA divergence between C57BL/6J and SPRET/EiJ, we sequenced two fibroblast cell lines with two

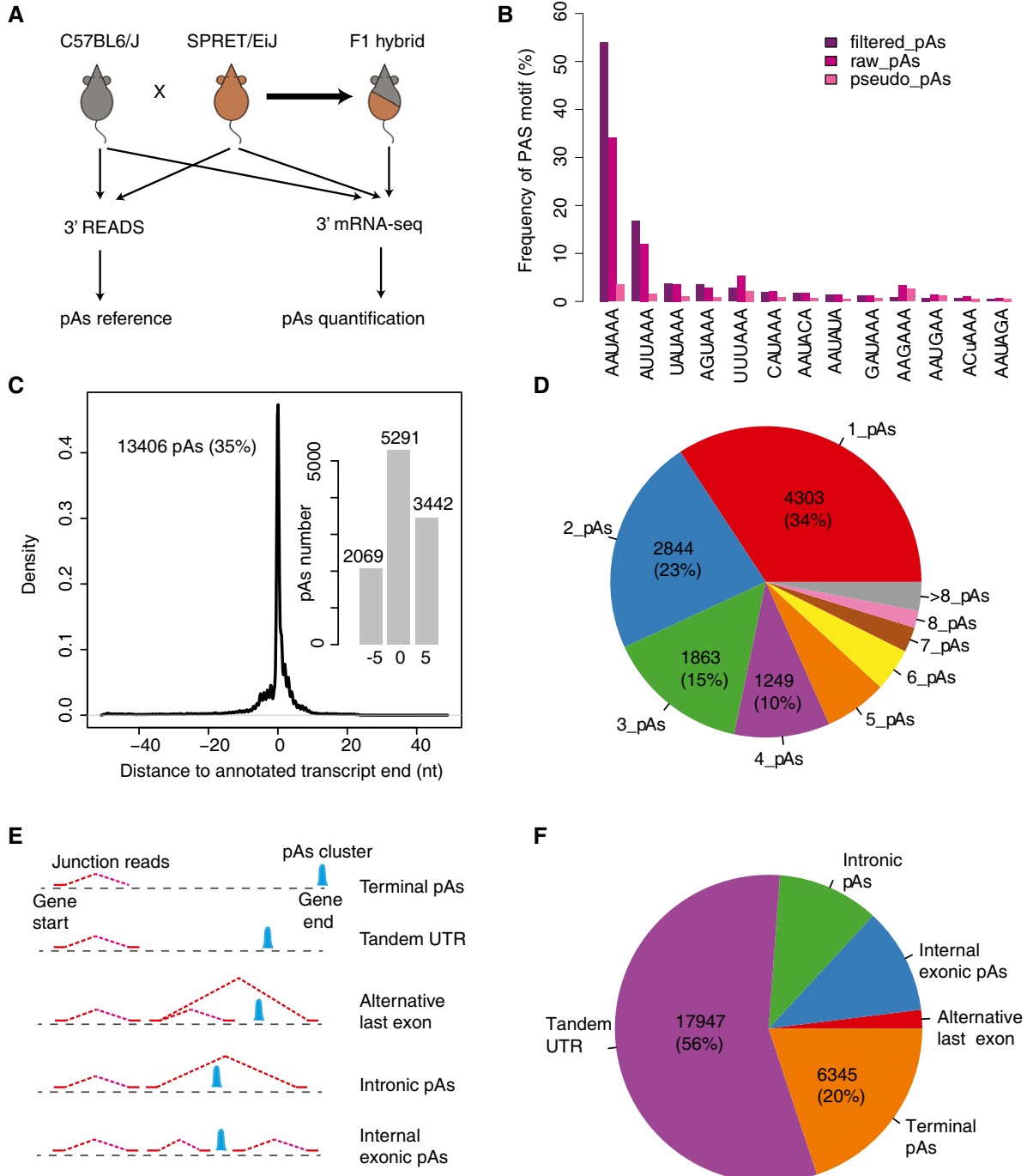

**Figure 1. Experimental design and construction of the pAs reference.**

A    Study design. 3′ READS method was used to identify the pAs expressed in the fibroblasts from C57BL/6J and SPRET/EiJ mouse strains. 3′ mRNA-Seq method was used to quantify the usage of the identified pAs in both two parental strains and their F1 hybrids.

B    The frequencies of 13 known PAS motifs in the 100 nt upstream region of pAs identified in C57BL/6J. pseudo_pAs represents pAs identified by using reads without non-genomic T (Materials and Methods). raw_pAs and filtered_pAs represent the pAs determined by the PASS reads with and without further filtering, respectively (Materials and Methods). X-axis shows different types of PAS motifs, and y-axis shows the percentage of pAs with the specific motif in the upstream region. See Fig EV1C for the results from SPRET/EiJ.

C    13,406 pAs identified in this study were located within 50 nt away from the 3′ end of RNA transcripts annotated in Ensembl. X-axis shows the distance between the 13,406 pAs identified in this study and the annotated pAs. Y-axis represents the density. Inset: Barplot shows the number of pAs with representative cleavage site identical to the annotated transcript ends as well as those within 5 nt upstream or downstream to the annotated ends, respectively.

D    The pie chart shows the percentage of protein-coding genes with different number of pAs.

E    Schematic definition for different types of pAs. See Materials and Methods for the details.

F    Pie chart shows the percentage of different types of pAs identified APA in this study.

replicates by 3′ mRNA-Seq on Illumina HiSeq 2000/2500 platform (Fig 1A; Materials and Methods). On average, 58.6 million reads were generated from each parental sample, of which 57.5 million were retained after trimming off the adaptor and random primer sequences. Among these filtered reads, around 60% was uniquely mapped to either C57BL/6J or SPRET/EiJ genome using Tophat2 (Table EV2), of which about 69% could be mapped to the reference pAs clusters and were used to quantify pAs usage. In the fibroblast cells derived from both parental strains, the estimated pAs expression was highly correlated between the two replicates, demonstrating the good reproducibility of 3′ mRNA-Seq ($r = 0.98$; Fig EV2A and B). Moreover, gene expression level quantified by counting all the 3′ mRNA-Seq reads mapped to the pAs clusters of each gene correlated well with that by standard mRNA-Seq, which further demonstrated its quantitative nature (Fig EV2C–F).

We next used DEXSeq to assess the differential pAs usage between the two parental strains (Materials and Methods; Anders *et al*, 2012). To ensure high accuracy, we considered only autosomal protein-coding genes with at least two identified pAs, which were expressed ($\geq 20$ pAs reads) in both fibroblasts and of which each alternative pAs cluster was supported by at least 10 reads from the two strains together (Materials and Methods). Based on these criteria, out of 24,721 pAs in 7,271 genes, we identified in total 3,747 pAs showing significant divergence between the two parental strains (Benjamini–Hochberg-adjusted *P*-value < 0.01, delta percentage of pAs usage > 10%; Table 1). Figure 2A showed two representative examples with significant pAs divergence, biased toward C57BL/6J and SPRET/EiJ strains, respectively. Using an independent 3′ RACE-based approach (Materials and Methods), we further validated the biased expression of their APA isoforms. More importantly, the difference in the relative isoform abundance between the two strains agreed to that determined by the global approach (Fig 2A).

These divergent events covered all the five different categories, with the higher frequency in the last exon including both terminal and tandem 3′ UTR pAs (Table 1). APA can affect either protein-coding and/or only 3′ UTR. The former might be subject to stronger

selection during evolution. Consistent with this, among the divergent pAs, the frequency of divergent pAs affecting only 3′ UTR was significantly higher than that involving coding region, demonstrating that in general APA with functional relevance was under stronger negative selection (Table 1).

## Predominant contribution of *cis*-regulatory variants underlying APA divergence between C57BL/6J and SPRET/EiJ

Differential pAs usage between species could arise from *cis*- and/or *trans*-regulatory divergence. After identifying the differences between the two parental strains, we next addressed the relative contributions of *cis*-regulatory effects on the APA divergence using their F1 hybrids. *Trans*-acting contributions can then be inferred by comparing the differences between the two alleles in the F1 hybrids to those between the parental strains.

We sequenced the RNA extracted from F1 fibroblast cell line using 3′ mRNA-Seq approach and obtained an average of 230 million reads for each of the two replicates. After trimming off adaptor and random primer sequence, about 143 million reads were uniquely mapped to either C57BL/6J or SPRET/EiJ genome for each replicate. The high density of sequence variants between the genomes of C57BL/6J and SPRET/EiJ allowed the unambiguous assignment of allelic origin for on average 67.3 million reads, which were used for further quantification of allelic pAs usage (Table EV2).

To avoid the potentially inaccurate pAs quantification due to lower percentage of allele-specific reads, we first created a mock F1 hybrid 3′ mRNA-Seq dataset by mixing reads derived from the two parental strains. We then compared the number of allele-specific reads to that from the original parental dataset for both strains. In total, 12,809 pAs clusters with lower percentage of allelic reads were filtered out (Materials and Methods). Out of the remaining 16,998 clusters from the protein-coding genes expressing multiple pAs, 12,749 pAs clusters supported with sufficient allelic reads for each replicates of the F1 hybrids were retained for further analysis (Materials and Methods). The allelic differences of pAs usage measured in the two replicates were highly correlated ($r = 0.90$; Fig EV3). After applying the same approach as that for parental strains, we detected a total of 2,618 divergent pAs usage between the two alleles in F1 hybrids (Benjamini–Hochberg-adjusted *P*-value < 0.01, delta percentage of pAs usage > 10%).

To assess the accuracy of our allele-specific APA analysis, we selected 20 candidate genes for validation. Using PacBio RS system, we deep sequenced the 3′ RACE products amplified from F1 hybrid fibroblasts using primers targeted at pAs upstream regions with no sequence variant between the two strains (Fig 2A; Materials and Methods; Eid *et al*, 2009; Sun *et al*, 2013). The PacBio reads were assigned to different isoforms from the two alleles, which were then counted to calculate the allele-specific pAs usage. As shown in Fig 2B, the difference in APA patterns between the two alleles estimated in this way were highly correlated with those determined by 3′ mRNA-Seq ($r = 0.98$).

We then compared the divergent pAs usage between the two alleles to that between the two parental strains. Out of 11,306 pAs clusters retained for both strain- and allele-specific APA analysis, 2,532 had divergent regulation between the parental strains, of which 1,876 and 572 exhibited *cis*- and *trans*-divergence, respectively (Fig 2C; Materials and Methods). Such predominant

**Table 1. Comparison of alternative polyadenylation between C57BL/6J and SPRET/EiJ.**

| | No. of total expressed pAs | No. of divergent pAs (%) | *P*-value (Fisher's exact test) |
|---|---|---|---|
| **pAs types** | | | |
| Alternative last exon | 542 | 80 (14.8) | |
| Internal exon | 2,196 | 240 (10.9) | |
| Intronic | 2,363 | 270 (11.4) | |
| Tandem 3′ UTR | 13,874 | 2,029 (14.6) | |
| Terminal | 5,746 | 1,128 (19.6) | |
| **Effect** | | | |
| Coding regions[a] | 5,101 | 590 (11.6) | $9.9e^{-13}$ |
| Non-coding regions[b] | 19,620 | 3,157 (16.1) | |

[a]Including Alternative last exon, Internal exon, and Intronic pAs.
[b]Including Tandem 3′ UTR and Terminal pAs.

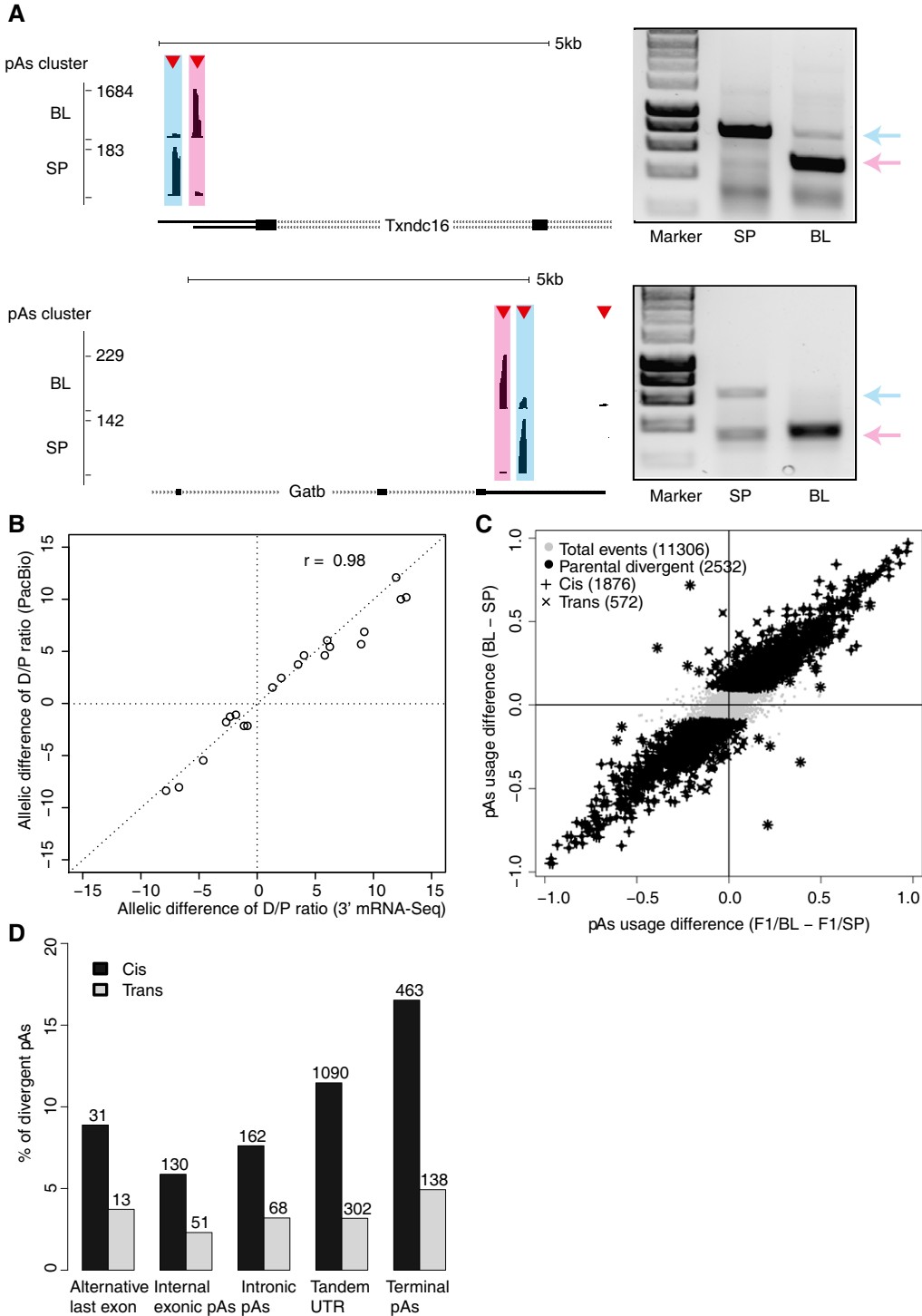

**Figure 2.  Dissection of *cis*- and *trans*-contribution in APA divergence.**

A   Two representative examples of parental divergent pAs events in gene *Txndc16* (up) and *Gatb* (down). In the left panel, above each gene structure, the red triangles represent the identified pAs clusters and the black bars represent 3′ mRNA-Seq reads mapped within distinct pAs clusters. Pink and blue shades mark proximal and distal pAs, respectively. The right panel shows the independent validation of the two divergent pAs using 3′ RACE method. The gel images illustrate 3′ RACE products obtained from C57BL/6J (BL) and SPRET/EiJ (SP) fibroblasts. Positions of the distal and the proximal pAs isoforms are indicated with blue and pink arrows, respectively.

B   Scatterplot showing the comparison of allelic difference (log2 C57BL/6J (BL) and SPRET/EiJ (SP)) in distal/proximal pAs usage ratio estimated by 3′ mRNA-Seq (*x*-axis) and that by PacBio sequencing (*y*-axis). Each circle represents one gene, and in total, 20 candidate genes were randomly chosen for validation.

C   Scatterplot comparing the parental difference in pAs usage to the allelic difference in F1 hybrids (*x*-axis). Out of 2,532 divergent APA events between the parental strains, 1,876 (indicated as "+") and 572 (indicated as "x") exhibit significant *cis*- and *trans*-regulatory divergence, respectively.

D   Percentage of the five types of pAs events regulated by *cis*- and *trans*-divergence (numbers of events for each type are labeled above the bars).

*cis*-contribution was evident for all the five different types of pAs (Fig 2D).

### Sequence variants in the pAs flanking regions associated with *cis*-divergent pAs

*Cis*-regulatory divergence should result solely from variants in genomic sequences, particularly those residing close to the cleavage sites. To investigate this, we calculated the frequencies of sequence variants in the regions (−100 nt to 100 nt) flanking the cleavage sites of pAs with or without *cis*-regulatory divergence. As shown in Fig 3A, compared to the non-divergent control events (Materials and Methods), the flanking regions of the *cis*-divergent pAs contained significantly higher density of sequence variants between the two strains.

According to previous studies, the flanking region could be separated into four windows, which contain potentially different *cis*-regulatory elements: auxiliary upstream elements (AUE) [−100, −40), core upstream elements (CUE) [−40, 0), core downstream elements (CDE) (0, 40], and auxiliary downstream elements (ADE) (40, 100] (Hu *et al*, 2005). To explore how the sequence variants were distributed in the four different windows, we calculated and compared the variant density for each window separately. As shown in Fig 3B, compared to 2,502 control pAs, the *cis*-divergent pAs showed higher frequency of sequence variants more evidently in the upstream and downstream proximal region containing potential CUEs and CDEs, respectively. Consistent with this, when we further determined the variant density in the flanking region with higher spatial resolution (Materials and Methods), compared to the control pAs, the pAs with *cis*-divergence exhibited significantly higher variant enrichment in the two proximal regions (Fig 3C). Interestingly, the difference of variant density between divergent and control pAs is most prominent in the narrow window (−30 nt to −15 nt) known to contain the PAS motifs, that is AAUAAA and its close variants.

To further examine whether the sequence difference in the flanking regions are responsible for the observed pAs divergence,

we used a reporter assay system (Materials and Methods). In brief, we employed the pRIG vector, which could produce two transcript isoforms when a pAs is inserted into the multiple cloning site (MCS; Pan *et al*, 2006; Ji *et al*, 2009). To compare usage of the pAs flanked by the genomic sequences derived from the two alleles, the flanking sequences from the two alleles were separately cloned into the vector and transfected into 3T3 cells. Quantitative real-time PCR (qRT–PCR) was carried out with primers targeting the region either upstream (RFP) or downstream of the inserted pAs. The relative pAs strength could then be estimated as the ratio of mRNA abundance between RFP and EGFP. As shown in Fig 3D, 10 allele pairs that were tested showed significantly differential pAs usage biased toward the same alleles as observed in our global analysis. These results demonstrated that sequence variants in the pAs flanking regions indeed confer significant contribution to the observed divergence in pAs usage.

### RNA secondary structure in the upstream proximal region inhibits pAs usage

RNA secondary structure is known to influence almost every step in the process of gene expression, such as splicing, miRNA targeting, and translation (Tuller *et al*, 2010; Jin *et al*, 2011; Gu *et al*, 2014; Mao *et al*, 2014; Wang *et al*, 2016). However, its effect on pAs regulation is still not clear. To explore whether and how RNA secondary structure affect pAs usage, we compared the minimum free energy (MFE) of mRNA segments in the four windows flanking the cleavage sites between the two alleles (Materials and Methods), and correlated such difference to the observed allelic divergence in pAs usage. Interestingly, only in the upstream proximal region (containing CUEs), the alleles with less stable local secondary structure were more likely to have higher pAs usage (Figs 3E and EV4A–C), indicating the formation of stable secondary structure in upstream proximal region inhibited the pAs usage.

A recent genome-wide analysis of RNA secondary structure in human lymphoblastoid cells has demonstrated that RNA fragments

---

**Figure 3.  Sequence features associated with *cis*-divergent pAs usage.**

A   The cumulative distribution function (CDF) of sequence variant density (number of variants per base) in the pAs flanking regions (−100 nt to 100 nt) for the events with *cis*-regulatory divergence (black) and without (control, gray). Compared to controls, the events with significant *cis*-regulatory impact have higher sequence divergence in the flanking regions. The *P*-value was calculated using Mann–Whitney *U*-test.

B   Barplot showing the sequence variant density in the four windows flanking pAs for those with *cis*-regulatory divergence (black) and without (control, gray) separately. According to previous publications, the pAs flanking region was separated into four windows, which contain potentially different *cis*-regulatory elements: auxiliary upstream elements (AUE) [−100, −40), core upstream elements (CUE) [−40, 0), core downstream elements (CDE) (0, 40], and auxiliary downstream elements (ADE) (40, 100]. Compared to the control pAs, the *cis*-divergent pAs show the higher frequency of sequence variants more evidently in the upstream and downstream proximal region containing potential CUEs and CDEs, respectively.

C   High-resolution comparison of sequence variant density in the pAs flanking regions between the pAs with significant *cis*-divergence (black) and without (control, gray). To determine the variant density in the flanking region (−100 nt to 100 nt) with higher spatial resolution, we calculated the density of sequence variants in an 8 nt sliding window with the step size of 1 nt. Blue line represents the difference of variant density between *cis*-divergent and control pAs events. Red stars indicate the statistical significance (Bonferroni-adjusted *P*-value < 0.05), which was determined by Mann–Whitney *U*-test. Compared to the control pAs, the pAs with *cis*-divergence exhibit significantly higher variant density in the two proximal regions. The difference of variant density is most prominent in the narrow window (−30 nt to −15 nt, marked in red shade) known to contain the PAS motifs, that is AAUAAA and its close variants.

D   Allele-specific sequences in the pAs flanking regions are able to drive the observed allele-specific pAs usage. An *in vitro* reporter assay system was used to compare the usage of pAs flanked by the genomic sequences derived from the two alleles (Materials and Methods). The relative pAs usage was calculated by the ratio of RFP to EGFP mRNA abundance. Ten pAs, five with usage biased toward SPRET/EiJ allele (left to the dash line) and five with usage biased toward C57BL/6J allele (right to the dash line) were tested. All the 10 reporter pairs show significant differential pAs usage biased toward the same allele as observed in our global analysis (*n* = 3; mean ± SEM; **P* < 0.05; ****P* < 0.001; Student's *t*-test).

E   The cumulative distribution function (CDF) of allelic difference in MFE of core upstream elements (CUE) from pAs with usage biased toward C57BL/6J (BL > SP, blue line), SPRET/EiJ (SP > BL, red line), or without biases (control, gray). The statistical significance of the difference between the biased group and control group was determined by Kolmogorov–Smirnov test. Apparently, the alleles with less stable local secondary structure are more likely to have higher pAs usage.

       

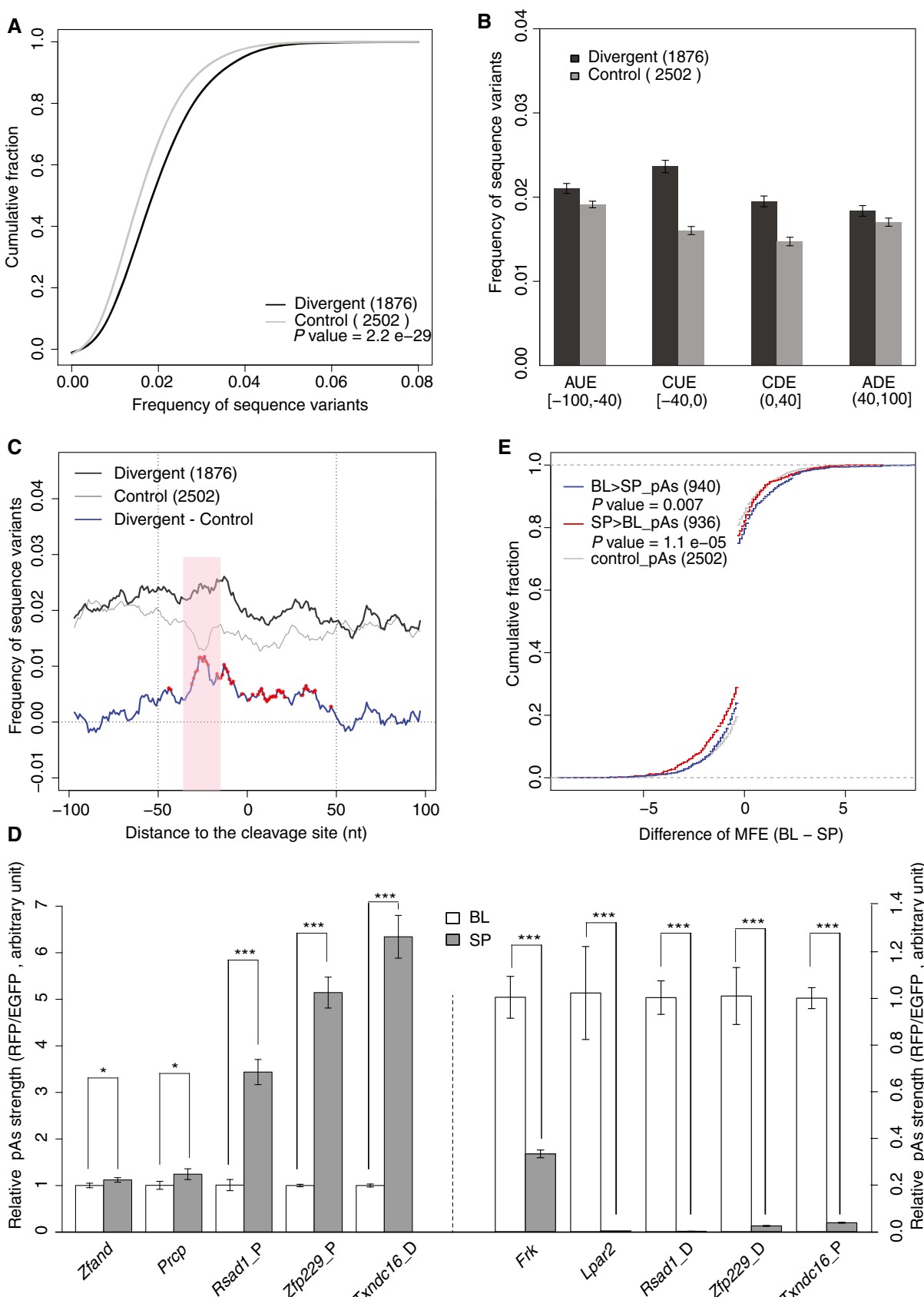

**Figure 3.**

in the vicinity of start and stop codons tend not to form stable secondary structure (Wan *et al*, 2014). To further examine the effect of RNA secondary structure on pAs usage, we took advantage of this high-quality, genome-wide dataset, and extended the analysis to the regions flanking the cleavage sites. As shown in Fig EV5, two regions, 15–30 nt upstream of the cleavage site and the cleavage site itself, showed a clear drop of PARS score, indicating the reduced tendency of double-stranded conformation and increased accessibility. This trend became more evident if we restricted our analysis to the annotated most distal pAs, which were in general of higher strength than proximal ones (Fig EV5). These, together, suggested that the regions immediate upstream of the cleavage sites, particularly for the pAs of high strength, need to be less structured, likely in order to guarantee the accessibility of the critical *cis*-elements.

## Sequence motifs associated with pAs strength

To further identify potential regulatory sequence elements, we correlated the allelic difference in the frequency of all hexamers within the region flanking the cleavage sites to the observed allelic pAs divergence (Materials and Methods). As shown in Fig 4A, in the upstream region, the canonic PAS motif AAUAAA displayed the highest positive correlation with the increased pAs usage, whereas UUUUUU is negatively correlated. To determine whether the negative correlation of UUUUUU with pAs usage is dependent on the presence of AAUAAA, we repeated the analysis after removing all the pAs containing AAUAAA in the upstream region. As shown in Fig EV6A, we could still observe the negative correlation of UUUUUU with pAs usage, and interestingly, the second most abundant PAS motif AUUAAA now displayed the highest positive correlation with the increased pAs usage. In contrast to the upstream region, in the downstream region, no hexamers were found to correlate with allelic difference in pAs usage (Fig EV6B). To exclude the possibility that a small number of pAs contributed excessively to the observed allelic difference in hexamer frequency, we assigned the allelic comparison of pAs usage into three groups, that is down-regulated, up-regulated, and unchanged (control), and counted the numbers of instances with the specific motif mutated or not mutated in the three groups, respectively. As

shown in Fig 4B and C, the down-regulated group showed significantly increased mutation rate than the other two groups for both AAUAAA and AUUAAA motifs. In contrast, compared to control group, UUUUUU were less likely mutated in the down-regulated group, but more likely mutated in the up-regulated group (Fig 4D). To further substantiate this finding and assess whether the presence/absence of these hexamers could predict the allelic pAs divergence, we separated the pAs into two groups, one containing these hexamers in both alleles or neither, and the other with the hexamers in only one allele, and compared the differences of allelic pAs usage between the two groups. As expected, mutation of AAUAAA, the canonical PAS motif, and AUUAAA, the most frequently used variant, significantly decreased pAs usage (Fig 4E and F). In contrast, the presence of UUUUUU inhibited the use of downstream pAs, although with much subtler effect (Fig 4G). Interestingly, not only a stretch of six Us, but also a stretch of seven or eight Us could confer similar inhibitory effect (Fig EV6C–E). Moreover, the more severe the disruption of the poly(U) stretches, the more substantial the inhibitory effect, further suggesting the functional role of poly(U) tract in the upstream region of the cleavage sites (Fig EV6C–E). Figure 4H showed one representative example for each of the three motifs, respectively. Encouraged by the success of this motif analysis, we applied a similar hexamer analysis also to the trans-regulated pAs. Here, we compared the frequency of all hexamers within 100nt upstream of the cleavage sites between trans-regulated and control pAs without parental divergence. However, no hexamers showed significantly biased frequency between the two groups (Fig EV6F).

To experimentally validate the effect of poly(U) tract on pAs usage, we chose one divergent pAs in *Alg10b* gene. In its upstream region, a tract of six continuous U was present only in the SPRET/EiJ allele, whereas four of the six Us were deleted in the C57BL/6J allele (Fig 4H, low panel). Using the same reporter assay as before, we first compared the usage of the pAs flanked by the genomic sequences derived from the two alleles (Materials and Methods). As shown in Fig 4I, the pAs usage was higher when the genomic sequence was derived from C57BL/6J allele. Then to directly determine the effect of poly(U) on pAs strength, we generated two additional constructs: The only change to the original vectors was at the poly(U) tract

---

**Figure 4.  Sequence motifs associated with pAs strength.**

A     Scatterplot comparing the allelic difference in hexamer frequency in upstream region (−100 nt to 0 nt) between two groups of pAs, one with usage biased toward C57BL/6J allele (BL, *x*-axis) and the other toward SPRET/EiJ allele (SP, *y*-axis), respectively. Each gray dot represents one specific hexamer, and the dot size represents the total frequency. The hexamers present in upper right quadrant are candidates for pAs enhancer, whereas the hexamers in lower left quadrant are candidates for pAs repressor. The canonic PAS motif AAUAAA displays the highest positive correlation with the increased pAs usage, whereas UUUUUU is negatively correlated.

B–D   The frequency of the sequence motif (B: AAUAAA; C: AUUAAA; D: UUUUUU) mutated in the three groups of the allelic comparison of pAs usage, that is usage decreased, increased, and unchanged. Pearson's chi-squared test was used to determine the statistical significance (*$P < 0.05$, ***$P < 0.001$).

E–G   Boxplots showing the allelic difference in pAs usage between the two pAs groups, one containing the sequence motif (E: AAUAAA; F: AUUAAA; G: UUUUUU) in both alleles or neither, and the other in only one allele. Mann–Whitney *U*-test was used to determine the statistical significance (*$P < 0.05$, ***$P < 0.001$). The solid horizontal bars, box ranges, the upper and lower bar represent median, 75th percentile, 25th percentile, maximum and minimum value, respectively.

H     Three representative examples show that the sequence motif mutated in one allele induces the decrease (upper: AAUAAA; middle: AUUAAA) or increase (lower: UUUUUU) of pAs usage. The black bar represents the 3′ mRNA-Seq read density. Red triangles represent the identified pAs cluster and the pink shade marks the pAs associated with the mutated motif. The motif sequences from both alleles are shown on the top of each panel, with the sequence variants in the motif marked in red.

I     *In vitro* reporter assays demonstrated the negative impact of poly(U) tract on pAs usage. Top: Schematic diagrams of reporter constructs for validating the impact of poly(U) tract on the usage of a pAs from *Alg10b* gene. The pAs flanking sequences from C57BL/6J (BL) and SPRET/EiJ (SP) alleles were inserted separately into the pRIG vector. In addition, two constructs with the only change to the original vectors being at the poly(U) tract (BL2SP and SP2BL) were generated. Low: The barplot shows the relative pAs strength for constructs with different inserts. Student's *t*-test was used to determine the statistical significance ($n = 3$; mean ± SEM; *$P < 0.05$; **$P < 0.01$). See Materials and Methods for the details on the cloning and quantification of pAs strength.

(Fig 4I, BL2SP and SP2BL; Materials and Methods). As shown in Fig 4I, the pAs usage decreased significantly upon restoring the UUUUUU element in C57BL/6J allele (BL2SP), while the pAs usage increased significantly by disrupting the intact UUUUUU in SPRET/EiJ allele (SP2BL; Fig 4I). These together unambiguously demonstrated the negative regulatory effect of poly(U) tract on pAs usage.

## Discussion

APA, which is regulated by both *cis*-elements and *trans*-factors, can affect transcriptome and proteome through generating mRNA isoforms that differ in coding and/or 3′ UTR regions, and thereby potentially affects the stability, translation, localization of target

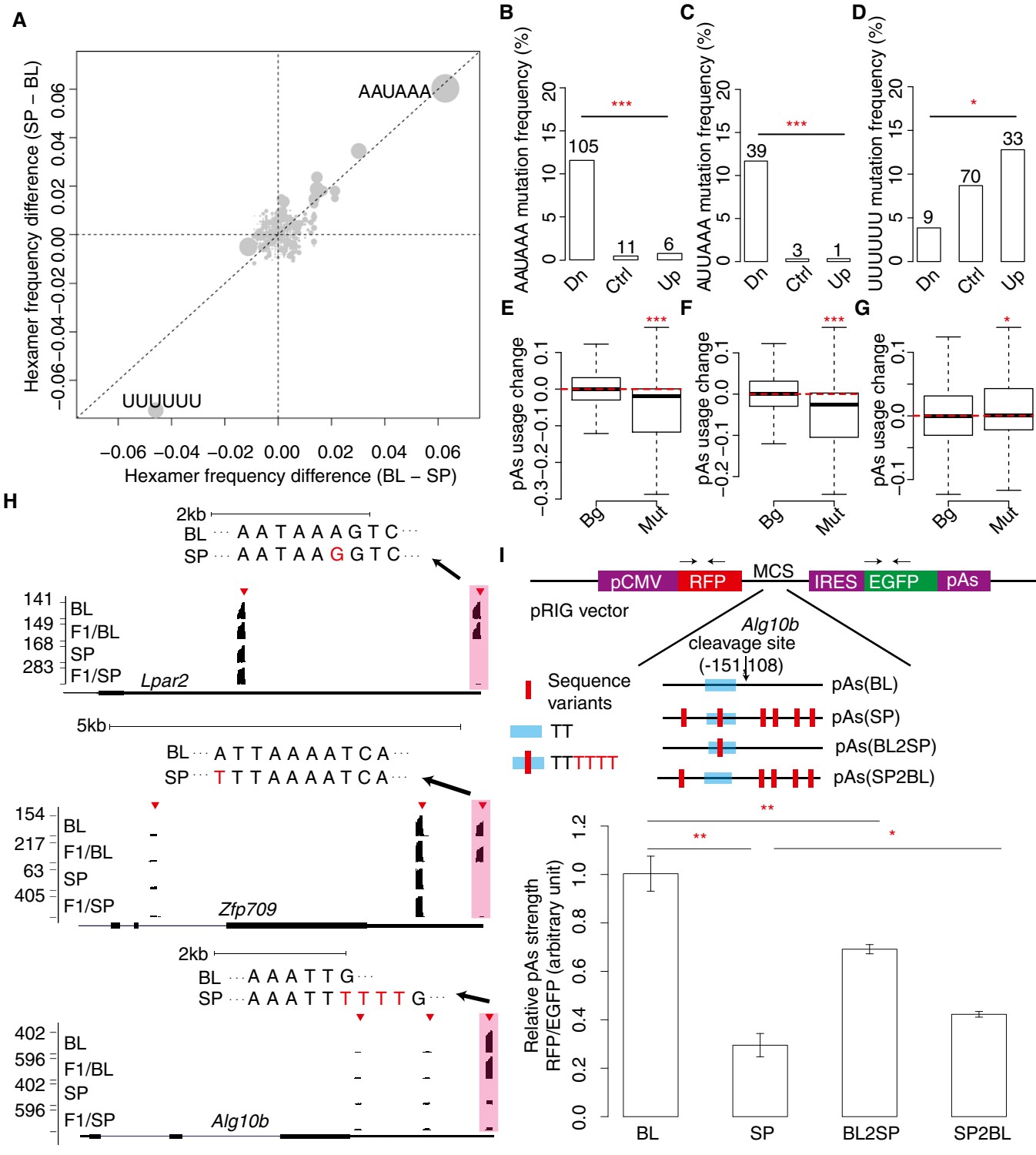

**Figure 4.**

mRNAs, and/or the function of resulted proteins (An *et al*, 2008; Loya *et al*, 2008; Mayr & Bartel, 2009; Elkon *et al*, 2013; Tian & Manley, 2013; Berkovits & Mayr, 2015). However, compared to other regulatory layers along the multi-step gene expression process, the extent of APA divergence during evolution remains largely unexplored, in particular how such divergence could arise from the change in *cis*-regulatory elements and/or *trans*-acting factors is totally unclear. In this study, to address these questions for the first time in a mammalian system, we globally identified the pAs and quantified their usage in two mouse strains as well as the allelic pAs usage in their F1 hybrids. In total, we identified 51,446 pAs for 13,808 genes expressed in the fibroblasts of the two mouse strains. Out of 24,721 pAs from 7,271 genes expressing multiple pAs, 3,747 pAs (15.2%) showed significant differential usage between the two strains. In comparison, only 796 (6.7%) alternative splicing (AS) events showed differential pattern between the fibroblasts of the same two strains (Gao *et al*, 2015). The higher divergence of APA on one hand indicates that in general APA confers less functional change to the gene function than AS, therefore under weaker negative selection. This is consistent with the observation that for both APA and AS, the alternative events affecting only non-coding regions displayed higher divergence than those affecting protein-coding sequences. On the other hand, the higher incidence of divergent APA implicates that modulation of pAs usage might be relatively easier to achieve. Therefore, during evolution, APA could provide fertile grounds for evolving new transcript isoforms with novel functions or even novel genes.

By comparing the allelic divergent pAs usage in F1 hybrids to those between the parental strains, we demonstrated clearly the predominant *cis*-contribution and such predominant *cis*-contribution were evident for all the five different types of pAs. The same holds also true for AS, as we observed before (Gao *et al*, 2015). The much higher prevalence of *cis*-divergence indicates at least for RNA processing including both APA and AS, the evolutionary changes with local effects were much more tolerated than those with pleiotropic consequences.

*Cis*-regulatory divergence should result solely from variants in genomic sequences. Indeed, compared to those non-divergent controls, the divergent pAs showed significantly higher density of sequence variants in the flanking regions, particularly in the proximal regions with the peak at the sites known to contain PAS motif. In the following analysis of sequence motifs, the mutation of canonical PAS motif AAUAAA and its closest variant AUUAAA appear to be the most prominent contributors to the observed *cis*-divergence in pAs usage. These indicate that the PAS signal, recognized by CPSF, is the most critical *cis*-elements for pAs definition.

The large set of pAs with *cis*-divergent usage collected in this study enabled to systematically characterize, apart from known PAS motifs, novel sequence features in APA regulation. By correlating the allelic difference in the stability of RNA secondary structure formed in the regions flanking the cleavage sites to the observed allelic divergence in pAs usage, we observed that the variants affecting local secondary structure in upstream proximal region could modulate pAs usage: The alleles with less stable secondary structure were more likely to have higher pAs usage. In consistent with this, by reanalyzing the previous genome-wide RNA secondary structure data obtained from human lymphoblastoid cells, we also found that the regions immediately upstream of the cleavage sites, especially for those of higher strength, were less structured.

Together, these observations suggest that, to enhance the pAs usage, this region needs to be freely accessible by the core CPSF machinery. We also noted that in another genome-wide analysis of RNA secondary structure in *Arabidopsis*, a structured–unstructured pattern was observed around the alternative polyadenylation sites: Whereas the cleavage sites themselves (−1 nt to 5 nt) were significantly less structured, RNA secondary structure upstream of the cleavage site from −15 nt to −2 nt showed higher stability (Ding *et al*, 2014). Several reasons could explain why their observation at the upstream region did not fully agree to ours. First, the studies were performed in different systems (*Arabidopsis* versus mouse or human) and the different observation may reflect the difference in APA regulation between plants and animals. Second, the RNA structures were analyzed by different methods. On one hand, our computational prediction could not always faithfully capture the *in vivo* status and in addition have limited spatial resolution. On the other hand, the two experimental methods used by Chang laboratory and Assmann laboratory were based on different strategies (differential activity of RNase with single-strand or double-strand specificities versus DMS reactivity). Last, more importantly, the *Arabidopsis* study restricted their analysis to the alternative polyadenylation sites, which might be of lower strength than the terminal ones. Indeed, as shown in Fig EV5, the RNA secondary structure pattern we observed in human lymphoblastoid cells was much more evident for the annotated most distal pAs. It awaits future studies on APA regulation to resolve the impact of RNA secondary structure and the underlying mechanisms.

In addition to the secondary structure, we also identified a poly(U) element in the upstream region of cleavage site, which exerts its function as a potential pAs repressor. The negative effect was validated using an exogenous reporter system. During the preparation of our manuscript, Zavolan laboratory published a paper that identified a poly(U) motif enriched in the vicinity of cleavage sites (Gruber *et al*, 2016). Their further functional study revealed that the heterogeneous ribonucleoprotein C (HNRNPC) could bind to this motif and repressed the usage of nearby pAs. However, the underlying molecular mechanisms and whether other poly(U) binding RBPs could also confer regulatory effect on pAs usage await for further investigations.

## Materials and Methods

### Cell culture and RNA extraction

The fibroblast cell lines from two parental mouse strains and the F1 hybrids used here were the same as in our previous studies (Gao *et al*, 2013, 2015; Hou *et al*, 2015). Cells were cultured in RPMI 1640 medium (Gibco, Life Technologies) with 10% FBS and 1% P/S. Total RNA were extracted from the cultured cells using TRIzol reagent according to the manufacture's protocol (Life Technologies). The integrity of total RNA was estimated by using Agilent Bioanalyzer with RNA Nano kit (Agilent Technologies), and RNA integrity number (RIN) above 9.0 was used for subsequent experiments.

### 3′ region extraction and deep sequencing (3′ READS)

To construct a reference pAs dataset for the two parental strains, the 3′ READS method previously published by Hoque *et al* (2013)

was utilized with some minor modifications. In brief, 30 μg total RNA was subjected to poly(A) selection with 200 μl Ambion oligo $(dT)_{25}$ beads (Ambion). Poly(A) containing RNA was then digested by 4 U RNase III (NEB) in the volume of 50 μl reaction system at 37°C for 40 min. After discarding the supernatant and the RNA-bounded beads were washed twice with binding buffer, the short poly(A) containing RNA was eluted by 100 μl elution buffer. Following binding $rU_5T_{45}$ oligos with biotin modification at 5′ end (IDT) to MyOne streptavidin C1 beads (Life Technologies), the eluted RNA was incubated with the oligo-coated beads in binding buffer for 1 h at room temperature on a rotor with gentle shaking to avoid the precipitation of beads and increase the binding efficiency. After second round of capturing, the supernatant was discarded and the beads were washed with stringent washing buffer, followed by 1 U RNase H (NEB) digestion in the volume of 50 μl for 30 min. Then the supernatant were collected and RNA was purified by phenol–chloroform extraction and ethanol purification. The pellet was dissolved in 8 μl ddH$_2$O.

Six microlitre of digested RNA was used for 3′ adaptor ligation with the truncated T4 RNA ligase II (NEB), followed by 5′ adaptor ligation with T4 RNA ligase I (NEB). The adaptor containing RNA was then reversed transcribed to cDNA using Superscript III Reverse Transcriptase (Life Technologies). The cDNAs were separated on 6% urea polyacrylamide gel, and only the range between 100 and 600 nt was exercised for gel extraction. Finally, the selected cDNA was amplified by indexed primers, which are compatible with Illumina sequencing. Ampure beads (Beckman Coulter) were used to purify the PCR products. The libraries were sequenced in single end 1 × 101 nt format on Illumina HiSeq 4000 platform. All the sequences of adaptors and primers are listed in Table EV3.

### 3′ mRNA sequencing

The usage of pAs was quantified by using the 3′ mRNA-Seq Library Prep Kit (Lexogen). Briefly, 500 ng total RNA was used as starting material. Poly(A) containing RNA was reverse transcribed by anchored oligo-dT primer, followed by second-strand cDNA synthesis with random primers containing part of the Illumina adaptor sequence. PCR amplification was then performed to get the Illumina sequencing library. The libraries were sequenced in single end 1 × 101 nt format on HiSeq 2000/2500 platform. The detailed information about the adaptors and primers could be found on the website of Lexogen (https://www.lexogen.com).

### Alignment of sequencing reads

The reference genome (mm10) and Ensembl gene annotation (GRCm38, release 74) of C57BL/6J were downloaded from Ensembl (http://www.ensembl.org), while the SPRET/EiJ reference genome and gene annotation were created as described in our previous paper (Gao *et al*, 2015).

For the 3′ READS reads, the leading T and tailed adaptor sequence were removed by using customized scripts and Cutadapt (http://cutada pt.readthedocs.io/en/stable/guide.html). Only the reads not shorter than 15 nt were aligned to the reference genomes using Tophat2 with the default parameters (Kim *et al*, 2013). Only the uniquely mapped reads with MAPQ ≥ 10 were retained for further analysis.

For 3′ mRNA-Seq reads, after trimming off the adaptor sequence at 3′ end, another 12 nt from the 3′ end derived from the random primer, which is used during the second-strand cDNA synthesis, was removed. After further removing the first nucleotide from the 5′ end, only the reads not shorter than 15 nt were aligned to the reference genomes by Tophat2 with default parameters (Kim *et al*, 2013).

For the sequencing data from parental strains, reads were aligned to the corresponding genome references, whereas for the real as well as mock F1 hybrid datasets, reads were aligned to both genomes and then assigned to parental alleles with smaller mapping edit distance. Reads with identical edit distance to both parental alleles were discarded, and the remaining allele-specific reads were retained for further analysis.

### pAs cluster identification

The uniquely mapped reads from 3′ READS data were compared with the genome sequences. Only the reads with at least two non-genomic T at 5′ end (2 T; PASS reads) were retained to call pAs clusters using the method as described in the previous studies (Tian *et al*, 2005; Hoque *et al*, 2013). In brief, 3′ end of the read alignments represented the cleavage sites and the cleavage sites located within 24 nt from each other were clustered together. If the cluster size was not larger than 24 nt, it was considered as one pAs cluster and the position with the highest read coverage was defined as the representative position for the cleavage site. If a cluster was larger than 24 nt, the position with the highest read coverage will be chosen as representative cleavage site and the remaining reads located at least 24 nt away from the position were re-clustered as described above. This process was repeated until all PASS reads were assigned to one cluster.

We then determined the potential false positives of the identified pAs. First, we estimated the enrichment of PASS reads within each pAs cluster using our previously published standard poly(A) RNA-Seq data (Gao *et al*, 2015) as background. In brief, for each pAs cluster, we counted two numbers, that is (i) PASS read count within the cluster (R), (ii) the ratio of the PASS read coverage (reads count/cluster size) within the identified pAs cluster to the RNA-Seq read coverage (reads count/window size) in the region (−500 nt to 50 nt) flanking the representative cleavage site (E). Second, we repeated the pAs calling using the reads without non-genomic T at 5′ end (0T) and then determined R- and E-values for each of these peusdo-pAs cluster as described above. Then, for each of the real pAs cluster, we determined a *P*-value as following:

In a total of N clusters, given any cluster $i$ ($R_i$, $E_i$), there would be $n_i$ clusters with both higher R and E, its P was calculated as:

$$P_i = n_i/N$$

For each pAs cluster, we calculated the *P*-value based on the real pAs cluster dataset ($P_i$ (2T)) and peusdo-pAs dataset ($P_i$ (0T)), the latter representing the false positive (FP). To balance sample sizes of real pAs and peusdo-pAs dataset, we subsampled the peusdo-pAs dataset with identical size as the real one for 100 times. The average of $P_i$ (0T; E ($P_i$ (0T))) was used to estimate Q-value as below:

$$Q_i = FP/(FP + TP) = E(P_i(0T))/P_i(2T)$$

Finally, based on our replicate data, we performed IDR analysis to retain only the reproducible pAs clusters (Li *et al*, 2011). Here, IDR

analysis was based on the *Q*-value determined in each of the two replicates. Only the pAs clusters with IDR value < 0.05 were considered as confident pAs clusters. Given the heterogeneity of cluster size and in order to make the following 3′ mRNA-Seq-based pAs quantification for each pAs equivalently, we redefined the pAs cluster as a window of 48 nt in length flanking the cleavage site (−24 nt to 24 nt).

## Filtering of C57BL/6J and SPRET/EiJ pAs clusters

Given the potentially incomplete/incorrect assembly of SPRET/EiJ genome reference sequences, some of the pAs identified in one strain might not be unambiguously mapped in the genome references of the other strain, for which it was infeasible to directly compare the usage between the two strains/alleles. Therefore, we retained only the pAs clusters, of which the region (−124 nt to 24 nt) flanking their cleavage sites could be reciprocally aligned between the two strains using LiftOver (https://genome.ucsc.edu/cgi-bin/hgLiftOver).

## pAs annotation

According to the Ensembl gene annotation, we assigned each pAs cluster to protein-coding genes, lincRNAs, other ncRNAs, as well as the intergenic regions. For the pAs assigned to protein-coding genes, we further classified the pAs into five categories based on Ensembl annotation as well as the RNA sequencing data, including terminal pAs (annotated Ensembl gene end), tandem 3′ UTR pAs, alternative last exon pAs, intronic pAs, and internal exonic pAs (Fig 1E).

## pAs usage quantification

The 3′ mRNA-Seq data were used to quantify pAs usage by counting the number of reads with the 5′ end located within the reference pAs cluster. If a protein-coding gene expressed ≥ 2 pAs in either strains/alleles (i.e., the strain/allele average read counts from each of these pAs ≥ 5; and the sum of the reads from all these pAs within one strain/allele ≥ 20), we compared the strain or allelic difference in the usage of each expressed pAs within the gene using DEXSeq (Anders *et al*, 2012).

## Further pAs cluster filtering with mock F1 hybrid data

In F1 hybrids, only the reads unambiguously assigned to specific alleles were kept for the quantification of pAs usage. Thus the region covered with lower percentage of allelic-specific reads could potentially lead to the inaccurate calculation of pAs usage difference between the two alleles. In order to reduce such potential errors, we created a mock F1 hybrid dataset by pooling the 3′ mRNA-Seq reads from the two parental strains together and then performed the same mapping procedure as that for the real F1 hybrid dataset. We assumed the assignment of each read to its corresponding allele was a Bernoulli trial. Therefore, for a pAs cluster with n reads mapped in the parental strain dataset, the number of reads assigned to the same allele in F1 mock dataset followed a binomial distribution B (n, p). The $P$ was estimated from the genome-wide average proportion of reads could be unambiguously assigned to the allele ($P$ = Total_mock/Total_parental, where Total_parental represents the number of all reads from all reference pAs mapped in the

original parental strain data and Total_mock represents those that could be unambiguously assigned to the allele). We filtered out the pAs that did not follow the binomial distribution (single tail binomial test, $P < 0.05$) or with $Reads_{parental} < Reads_{mock}$ ($Reads_{parental}$ represents the number of reads assigned in parental strain data and $Reads_{mock}$ represents the number of reads that could be unambiguously assigned to the allele) for either allele, and the remaining ones were retained for allelic pAs usage quantification in F1 hybrids.

## PacBio sequencing and data analysis

Total RNA was extracted from the F1 fibroblast cell line using TRIzol reagent (Life Technologies). Anchored oligo (dT) primers linked with common sequence were used for reverse transcription by superscript III (Life Technologies) following the manufacturer's protocol. Gene-specific primers targeting at the upstream region without sequence variants between C57BL/6J and SPRET/EiJ strains, together with primers targeting the common sequence, were used for PCR amplification. Two microlitre cDNA was used as template in a volume of 50 μl PCR system. PCR program was as follows: 5 min at 95°C; followed by five cycles of 30 s at 95°C, 30 s at 68°C/58°C, 30 s at 72°C and 30 cycles of 30 s at 95°C, 30 s at 65°C, 30 s at 72°C with 0.5°C reduction of Tm value for each cycle and a final extension at 72°C for 5 min; then hold at 4°C. The PCR products were purified by Agencourt Ampure XP system (Beckman Coulter)/gel extraction (Qiagen) and then quantified by Qubit HS dsDNA reagent (Life Technologies). Finally equal amount of each sample was mixed and sequenced on a PacBio RS SMART platform according to the manufacturer's instruction.

Sequencing data from the PacBio RS SMRT chip were processed through PacBio's SMRT-Portal analysis suite to generate circular consensus sequences (CCSs). The CCSs were then mapped to pAs flanking region of both C57BL/6J and SPRET/EiJ genome references using BLASTN with default parameters. Reads were assigned to a specific allele with better alignment score. The number of reads assigned to proximal and distal pAs of each allele was used to calculate the ratio of proximal to distal pAs usage for the two alleles and then compared to that determined by 3′ mRNA-Seq method. All primers used for RT and PCR were listed in Table EV4.

## Sequence variant density analysis

We calculated the density of sequence variants between the genomes of the two mouse strains in the regions (−100 nt to 100 nt) flanking the cleavage sites for both *cis*-divergent pAs and control pAs. The control pAs without *cis*-regulatory divergence were selected based on the following criteria: Benjamini–Hochberg-adjusted *P*-value > 0.5 and delta percentage of pAs usage < 5 %. The flanking region was separated into four windows: [−100, −40], [−40, 0), (0, 40], and (40, 100], which potentially contain different *cis*-elements according to previous study (Hu *et al*, 2005). The variant density was calculated and compared for each window separately. To further determine the variant density in the flanking region (−100 nt to 100 nt) with higher spatial resolution, we calculated the density of sequence variants in an 8 nt sliding window with the step size of 1 nt. The density of variants was compared between *cis*-divergent and controls pAs in each window by Mann–Whitney *U*-test, and the *P*-value was adjusted by "Bonferroni" method.

**Local RNA secondary structure**

Local RNA secondary structure minimum free energy (MFE) was calculated using RNAfold from the ViennaRNA package version 2.1.9 with default parameters at a temperature of 37°C (Lorenz *et al*, 2011). We compared the MFE of the four windows (as described above) flanking the cleavage site between the two alleles for the three pAs groups separately, that is pAs usage biased toward SPRETS/EiJ allele, pAs usage biased toward C57BL/6J allele, and pAs without allelic bias (controls). Kolmogorov–Smirnov test was used to assess the statistical significance of the difference between the biased group and control group.

Genome-wide dataset on RNA secondary structure across the human transcriptome (Wan *et al*, 2014) was downloaded from the sequence read archive (SRA) database at National Center for Biotechnology Information (accession numbers SRA100457). The sequencing reads were aligned to human reference genome (hg38) by using Tophat2 with default parameters. PARS scores were calculated on each base using the same method as described in the paper (Wan *et al*, 2014).

**Motif analysis**

To identify the motifs with regulatory effect on pAs usage, we first divided the *cis*-divergent pAs into two groups, (i) pAs with usage biased toward C57BL/6J allele, and (ii) pAs with usage biased toward SPRETS/EiJ allele. For each pAs from the two groups, we counted the hexamer occurrence in 100 nt upstream or 100 nt downstream of the cleavage sites from C57BL/6J and SPRETS/EiJ genomes, respectively. The allelic difference in hexamer frequency was then summed up and compared between the two groups. The hexamer with higher frequency in C57BL/6J genome for group 1 and with higher frequency in *SPRETS/EiJ* genome for group 2 was considered as pAs enhancer. In contrast, the hexamer with lower frequency in C57BL/6J genome for group 1 and with lower frequency in SPRETS/EiJ genome for group 2 was considered as pAs repressor. For the hexamer with potential significant impact on pAs usage including AAUAAA, AUUAAA, and UUUUUU, we calculated the allelic difference of pAs usage for those with only one allele containing the intact motif in the flanking region of the cleavage site and compared it to those with the motif in both alleles or neither.

**Vector construction and *in vitro* pAs reporter assay**

To investigate the effect of sequence difference in the flanking regions on pAs usage, we used the pRIG vector, which can generate two transcript isoforms when a pAs was inserted into the multiple cloning site (MCS; Fig 4I; Ji *et al*, 2009). The flanking regions of cleavage sites were amplified from the genomic DNA of both mouse strains separately. PCR products with homologous sequence at both ends were inserted into the pRIG vector using In-Fusion HD Cloning Kit (Clontech) following manufacturer's protocol. Sanger sequencing confirmed clones with correct inserts.

The 3T3 cells were maintained in DMEM (Gibco) with 10% FBS (Gibco). $3.0 \times 10^4$ cells were seeded in each well of 24-well plate 1 day before transfection. 500 ng plasmids (see above) were transfected to each well using 1 μl Lipofectamine 2000 reagent (Life Technologies) in three replicates according to the manufacturer's

protocol. After incubation for 24 h, the cells were harvested and lysed with TRIzol reagent (Life Technology) followed by extracting the total RNA using Direct-zol RNA kit (Zymo Research). To remove the potential remaining DNA, one microgram total RNA was treated with TURBO DNase (Ambion) in 20 μl reaction system following the manufacturer's protocol. Reverse transcription with oligo (dT) primers was then performed using Superscript III Reverse Transcriptase (Life Technologies). Quantitative real-time PCR (qRT–PCR) was carried out using SYBRGreen Masrermix I (Roche) on LightCycler 480 (Roche) with primers targeting RFP and EGFP regions in the vector, separately. The relative pAs strength was estimated as the RFP mRNA level normalized by the EGFP mRNA level, and compared between the two alleles.

To assess the potential effect of the UUUUUU motif, we chose one divergent pAs from *Alg10b* gene for further investigation. In brief, based on the plasmids constructed above, we created two additional ones by exchanging the UUUUUU sequence in C57BL/6J allele with UU in SPRETS/EiJ allele or vice versa (Fig 4I). To construct such plasmids, the Site-Directed Mutation System (Life Technology) was used and Sanger sequencing confirmed the mutants. After transfection, total RNA was extracted and treated with TURBO DNase followed with RT and qPCR. The procedures were the same as depicted above. Primers used for constructing the reporter vectors and qPCR were listed in Table EV5.

**Data access**

All the sequencing data generated from this study have been submitted to the European Nucleotide Archive (http://www.ebi.ac.uk/ena) under the accession number PRJEB15336.

**Expanded View** for this article is available online.

### Acknowledgements
We thank Dr. Jean Jaubert and Dr. Xavier Montagutelli from the Pasteur Institute for providing F1 hybrid mice. We thank Mirjam Feldkamp, Claudia Langnick, Claudia Quedenau, and Madlen Sohn for their excellent technical assistance. We thank Dr. Bin Tian from Rutgers New Jersey Medical School for kindly sharing of their pRIG vector. Mei-Sheng Xiao is supported by Chinese Scholarships Council (CSC).

### Author contributions
M-SX and WC conceived and designed the project. WS established the fibroblast cell lines. M-SX and Y-SL performed the experiments. BZ and M-SX analyzed the data with help from QG. M-SX, BZ, and WC wrote the manuscript. All authors read and approved the final manuscript.

### Conflict of interest
The authors declare that they have no conflict of interest.

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
