## [Review Process File · Molecular Systems Biology]

Global analysis of regulatory divergence in the evolution of mouse alternative polyadenylation

Mei-Sheng Xiao, Bin Zhang, Yi-Sheng Li, Qingsong Gao, Wei Sun and Wei Chen

Corresponding author: Wei Chen, Southern University of Science and Technology

Review timeline:

Submission date:	07 October 2016
Editorial Decision:	28 October 2016
Revision received:	08 November 2016
Editorial Decision:	11 November 2016
Revision received:	13 November 2016
Accepted:	15 November 2016

Editor: Maria Polychronidou

Transaction Report:

1st Editorial Decision

28 October 2016

Thank you again for submitting your work to Molecular Systems Biology. We have now heard back from the two referees who agreed to evaluate your study. As you will see below, the reviewers acknowledge that the presented analyses generate interesting insights. However, they list several issues, which we would ask you to address in a revision. The reviewers' recommendations are rather clear so I think that there is no need to repeat the points listed below, but please let me know in case you would like to discuss any specific point.

REFeree REPORTS

Reviewer #1:

The goal of the paper entitled "Global analysis of regulatory divergence in the evolution of mouse alternative polyadenylation" is to understand the degree of alternative polyadenylation (APA) divergence and the contributions of cis- and trans- regulatory elements to APA by applying F1 hybridization experiments into two evolutionarily distant mouse strains. Based on the sets of distinctly mapped reads in polyadenylation sites (pAs) from deep sequencing approaches in two strains, the authors provided corresponding experimental evidence for the following three observations. First, based on the frequency of divergent pAs in protein coding and non-coding regions, APA affecting their functions is more deleterious thus under strong negative selection. Secondly, through the comparison between two parental strains and their differences from the two alleles in the F1 hybrids, cis-effects are more dominant than trans-effects in APA. Lastly, both the stability of local RNA secondary structures and a poly(U) tract especially in the upstream region

have considerable effects on gene regulation based on the measurement of the minimum free energy (MFE) of mRNA segments and sequence motifs analysis, respectively. Overall, most results are relatively clearly explained and their experimental results are independently supported by using human genome-scale data. The authors also introduced a recently published paper showing different patterns of positional stabilities of RNA secondary structures in ADA in Arabidopsis and provided three probable scenarios/hypotheses explaining the observational disparity. In 2015, using similar approach, Chen and his colleagues have already published a paper in the same journal for the regulatory divergence in the evolution of alternative splicing. I think this paper can additionally provide more complete pictures of evolutionary history for post-transcriptional regulation in mouse.

Thus, I recommend that this paper be accepted for publication after some minor points explained below are addressed.

1. In the subsection, "Construction of the pAs reference", authors demonstrated the quality of their data by saying that most representative cleavage sites of the pAs clusters were almost identical to the annotated 3' end. This sounds somewhat subjective. It would be better to show more objective evidence such as quantitative measurements of their agreement. In the last sentence from the same paragraph, the authors should cite a paper showing "previous" observation.
2. Related to Fig2B, authors used 20 genes for validating the accuracy of their allele specific APA analysis. Authors need to mention that high replicability can be seen regardless of the choice of the selected genes and the numbers chosen.
3. In the subsection, "RNA secondary structure in the upstream proximal region inhibits pAs usage", authors said "This trend became more evident if we restricted our analysis to the annotated most distal pAs, which were in general of higher strength than proximal ones". Is there any figure or table which we can see these trends? If so, it should be referenced here.
4. In figure 1D and F, please add actual numbers on top of the percentages.

Reviewer #2:

In this manuscript Xiao et al. perform a global analysis of alternative polyadenylation (APA) using fibroblasts from two divergent mouse lines as well as their F1 cross. They combine data from 3'READS and oligo-dT priming based 3' quantification to annotate and measure the relative expression of APA. They focus their work on cis-regulated APA events and investigate potential motives contributing to its regulation. The authors perform orthogonal confirmation of selected targets using a fluorescence based in vitro system and analyze the contribution of secondary structure and motives to APA usage.

General remarks:

The combined use of two different 3' quantification methods allow the authors to focus on median and high expressed APA events and remove from their analysis any APA event due to internal oligo-dT priming. The fact that alterations of core polyadenylation elements (eg. hexamer AAUAAA) impacts APA are not surprising. However the authors use an elegant experimental design that allows them to distinguish between cis- and trans-regulated APA.

Major points:

Due to the experimental design used by the authors; I am surprised that they focus almost exclusively on the cis-regulated APA events. Adding a brief analysis of the trans-regulated APA events will significantly increase the interest of the paper and differentiate this work from other studies. For example, performing an hexamer analysis analogue to the one that the authors perform for the cis-regulated APA events. The authors could also study if different RNA Binding Proteins or miRNAs are putatively bound (or in proximity) to the alternative polyadenylated isoforms using available data (eg. PMID 23846655). And if so, analysis how is the expression of the putative RNA Binding Protein in the F1 cell line.

Minor points:

In page 12-13 the authors briefly mention the method that they use for orthogonal confirmation (eg. Fig 3D and 4I). However, the description in the main text is too brief. I would recommend adding a couple of sentences describing the general principle of the approach and how the artificial constructs are assayed in the same cell lines.

Some small typos in the figures (eg. in FigEV1D " Cleavage Site").

1st Revision - authors' response

08 November 2016

Text continued on next page.

Reviewer #1:

The goal of the paper entitled "Global analysis of regulatory divergence in the evolution of mouse alternative polyadenylation" is to understand the degree of alternative polyadenylation (APA) divergence and the contributions of cis- and trans-regulatory elements to APA by applying F1 hybridization experiments into two evolutionarily distant mouse strains. Based on the sets of distinctly mapped reads in polyadenylation sites (pAs) from deep sequencing approaches in two strains, the authors provided corresponding experimental evidence for the following three observations. First, based on the frequency of divergent pAs in protein coding and non-coding regions, APA affecting their functions is more deleterious thus under strong negative selection. Secondly, through the comparison between two parental strains and their differences from the two alleles in the F1 hybrids, cis-effects are more dominant than trans-effects in APA. Lastly, both the stability of local RNA secondary structures and a poly(U) tract especially in the upstream region have considerable effects on gene regulation based on the measurement of the minimum free energy (MFE) of mRNA segments and sequence motifs analysis, respectively. Overall, most results are relatively clearly explained and their experimental results are independently supported by using human genome-scale data. The authors also introduced a recently published paper showing different patterns of positional stabilities of RNA secondary structures in ADA in Arabidopsis and provided three probable scenarios/hypotheses explaining the observational disparity. In 2015, using similar approach, Chen and his colleagues have already published a paper in the same journal for the regulatory divergence in the evolution of alternative splicing. I think this paper can additionally provide more complete pictures of evolutionary history for post-transcriptional regulation in mouse. Thus, I recommend that this paper be accepted for publication after some minor points explained below are addressed.

R: We thank the reviewer for her/his positive comments on our study.

1. In the subsection, "Construction of the pAs reference", authors demonstrated the quality of their data by saying that most representative cleavage sites of the pAs clusters were almost identical to the annotated 3' end. This sounds somewhat subjective. It would be better to show more objective evidence such as quantitative measurements of their agreement.

R: We would like to thank the review to point this out. To make our statement more quantitative, we calculated the number of pAs with the identified representative cleavage site exactly identical to the ENSEMBL annotated transcript ends and those locating within 5nt upstream or downstream of the annotated ends, respectively. In the revised Fig 1C, we added an inset, which shows that 39.5% and 41.1% of these pAs are identical to or within 5nt upstream of downstream of the annotated ends, respectively. We also added these numbers in the revised main text (Page 7) and Figure legend (Page 38).

2. In the last sentence from the same paragraph, the authors should cite a paper showing "previous" observation.

R: We thank the reviewer for the suggestion and in the revised manuscript, we added the citation for the corresponding paper (Page 7).

3. Related to Fig2B, authors used 20 genes for validating the accuracy of their allele specific APA analysis. Authors need to mention that high replicability can be seen regardless of the choice of the selected genes and the numbers chosen.

R: Thank the reviewer for the suggestion. In the revised manuscript, to further assess the reproducibility of our method on measuring the allelic difference in pAs usage, we compared the results from the two independent experimental replicates. As shown in the newly added Fig EV3, we observed the results from the two replicated correlated well ($r = 0.90$).

4. In the subsection, "RNA secondary structure in the upstream proximal region inhibits pAs usage", authors said "This trend became more evident if we restricted our analysis to the annotated most distal pAs, which were in general of higher strength than proximal ones". Is there any figure or table which we can see these trends? If so, it should be referenced here.

R: We are sorry for the confusion. Actually we have showed the observation in Fig EV5, but forgot to cite in the text. As shown in Fig EV5, the red curve represents the level of RNA secondary structure around the annotated most distal pAs. In the updated manuscript, we cited the Fig EV5 at the end of this sentence (Page 14) and made it more clear at the figure legend as well (Page 45).

5. In figure 1D and F, please add actual numbers on top of the percentages.

R: We thank the reviewer for the suggestion. In the revised manuscript, we have added the actual numbers in Fig 1D and F.

Reviewer #2:

In this manuscript Xiao et al. perform a global analysis of alternative polyadenylation (APA) using fibroblasts from two divergent mouse lines as well as their F1 cross. They combine data from 3'READS and oligo-dT priming based 3' quantification to annotate and measure the relative expression of APA. They focus their work on cis-regulated APA events and investigate potential motives contributing to its regulation. The authors perform orthogonal confirmation of selected targets using a fluorescence based in vitro system and analyze the contribution of secondary structure and motives to APA usage.

General remarks:

The combined used of two different 3' quantification methods allow the authors to focus on median and high expressed APA events and remove from their analysis any APA event due to internal oligo-dT priming. The fact that alterations of core polyadenylation elements (eg. hexamer AAUAAA) impacts APA are not surprising. However the authors use an elegant experimental design that allows them to distinguish between cis- and trans-regulated APA.

R: We thank the reviewer for her/his positive comments on our study.

Major points:

Due to the experimental designed used by the authors; I am surprised that they focus almost exclusively on the cis-regulated APA events. Adding a brief analysis of the trans-regulated APA events will significantly increase the interest of the paper and differentiate this work from other studies. For example, performing an hexamer analysis analogue to the one that the authors perform for the cis-regulated APA events. The authors could also study if different RNA Binding Proteins or miRNAs are putatively bound (or in proximity) to the alternative polyadenylated isoforms using available data (eg. PMID 23846655). And if so, analysis how is the expression of the putative RNA Binding Protein in the F1 cell line.

R: We thank the reviewer for the important suggestion. Following this suggestion, we applied a similar hexamer analysis to those trans-regulated pAs. In brief, we compared the frequency of all hexamers within 100nt upstream of the cleavage sites between trans-regulated pAs and controls. The control pAs were selected based on the following criteria: 1) pAs should have a minimum expression level, i.e. BL + SP > 10 reads; 2) pAs need to have a minimum pAs usage, i.e. BL + SP > 10%; 3) in the comparison between two parental strains, Benjamini-Hochberg-adjusted P value > 0.5 and delta percentage of pAs usage < 0.05.

As shown in Fig R1 A, no hexamers shows significantly biased frequency between the two groups. Moreover, as the reviewer suggested, we also downloaded both the RBP binding motifs (PMID 23856655) and predicted miRNA binding sites (TargetScan), then compared their frequencies between the control pAs and trans-regulated pAs. Again, we failed to observe any motifs showing significant bias.

Fig R1: Scatterplot comparing the frequency of all hexamers (A) and RBP/miRNA binding sites (B) in the 100nt region upstream of cleavage sites between trans-regulatory pAs (X-axis) and control pAs (Y-axis).

Minor points:

In page 12-13 the authors briefly mention the method that they use for orthogonal confirmation (eg. Fig 3D and 4I). However, the description in the main text is too

brief. I would recommend adding a couple of sentences describing the general principle of the approach and how the artificial constructs are assayed in the same cell lines.

R: We thank the reviewer for the suggestion. In the revised manuscript, we described the general principle of the approach and how the artificial constructs are assayed in cell line (Page 13).

Some small typos in the figures (eg. in FigEV1D "Cleavage Site").

R: Thank the reviewer for helping us find the mistake. We have already checked and corrected the typos in Fig EV1D and other Figs as well.

2nd Editorial Decision

11 November 2016

Thank you again for submitting your work to Molecular Systems Biology. We have now evaluated the revised study and we think that the issues raised by the reviewers have been satisfactorily addressed. We would only ask you to include a couple of sentences in the main text referring to the analysis of trans-regulated pAs that was performed after the recommendation of reviewer #2.

2nd Revision - authors' response

13 November 2016

We thank you for your suggestion. Now, we added the analysis on trans-regulated pAs to the section “*Sequence motifs associated with pAs strength*” (Page 16) :

“Encouraged by the success of this motif analysis, we applied a similar hexamer analysis also to the trans-regulated pAs. Here, we compared the frequency of all hexamers within 100nt upstream of the cleavage sites between trans-regulated and control pAs without parental divergence. However, no hexamers showed significantly biased frequency between the two groups (Fig EV6F).”

In addition, the legend for Fig EV6F is also added on Page 46.

We hope that you find our revised manuscript now suitable for publication in Molecular Systems Biology.

3rd Editorial Decision

15 November 2016

Thank you again for sending us your revised manuscript. We are now satisfied with the modifications made and I am pleased to inform you that your paper has been accepted for publication.

Corresponding Author Name: Wei Chen

Manuscript Number: MSB-16-7375